# Effects of Multiple Planting Densities on Lignin Metabolism and Lodging Resistance of the Strip Intercropped Soybean Stem

**Bin Cheng** [1,2,†], **Ali Raza** [1,2,†], **Li Wang** [1,2], **Mei Xu** [1,2], **Junji Lu** [1,2], **Yang Gao** [1,2], **Sisi Qin** [1,2], **Yi Zhang** [1,2], **Irshan Ahmad** [1,2], **Tao Zhou** [1,2], **Bingxiao Wen** [1,2], **Wenyu Yang** [1,2] **and Weiguo Liu** [1,2,*]

[1] Key Laboratory of Crop Ecophysiology and Farming System in Southwest China, Ministry of Agriculture, Sichuan Agricultural University, Chengdu 611130, China; 2018201013@stu.sicau.edu.cn (B.C.); 2018601001@stu.sicau.edu.cn (A.R.); S20176512@stu.sicau.edu.cn (L.W.); 2018301013@stu.sicau.edu.cn (M.X.); 2019301099@stu.sicau.edu.cn (J.L.); 2019101006@stu.sicau.edu.cn (Y.G.); QinSisi@163.com (S.Q.); 2019201010@stu.sicau.edu.cn (Y.Z.); 2019605002@stu.sicau.edu.cn (I.A.); Zhoutao@cdutcm.edu.cn (T.Z.); wenbingxiao@outlook.com (B.W.); mssiyangwy@sicau.edu.cn (W.Y.)
[2] Institute of Ecological Agriculture, Sichuan Agricultural University, Chengdu 611130, China
[*] Correspondence: lwgsy@126.com; Tel.: +86-18628958371
[†] These authors contributed equally to this work.

**Abstract:** The shading of maize and self-shading are the key factors affecting the stem lignin biosynthesis and lodging resistance of soybean at middle and later growth stages in the strip intercropping system. A study was designed to explore the regulation mechanism of lignin metabolism and different planting densities; PD1, PD2, and PD3 were used having a total number of 17 plants m$^{-2}$, 20 plants m$^{-2}$, and 25 plants m$^{-2}$, respectively, on the lodging resistance of strip intercropped soybean stem. Our results depicted that the lower planting density (PD1) appropriately promoted the leaf photosynthesis activities (Pn), increase the activity of lignin-related enzymes and the accumulation of carbohydrates in stems, and eventually enhanced the lodging resistance of the strip intercropped soybean stem. Correlation analysis also showed that the lodging resistance index of soybean stem was significantly correlated with the available light for soybean canopy and Pn strip intercropped soybean stem characteristics and activities of enzymes related to lignin synthesis among the different planting densities. The findings of our research will be useful in future studies to understand the relationship between different light environment, planting densities, and lodging resistance of intercropped soybean and also guide the optimum planting density in maize–soybean intercropping system.

**Keywords:** maize–soybean strip intercropping; planting density; carbohydrates; lignin metabolism; lodging resistance index

## 1. Introduction

In the case of limited cultivated land resources in China, intercropping systems were important measures to improve land utilization efficiency [1–4]. Among intercropping systems, the maize–soybean intercropping system in southwest China has achieved the purpose of improving the yield of soybean and maize [5]. It is believed that the maize–soybean intercropping system has widely improved crop yield and nitrogen utilization efficiency via an increasing abundance of indigenous arbuscular mycorrhizal fungal (AMF) [6]. The maize–soybean intercropping system also achieved the purpose of saving fertilizer by promoting N uptake [7]. While these two crops were not sowing and harvesting at the same time, which was not conducive to mechanization. Regrettably, relay intercropped soybean

was subjected to shade stress of maize at the seedling stage, which resulted in soybean stem lodging easily [8]. However, the maize–soybean strip intercropping system could increase land utilization and mechanization efficiency, prevent continuous inter-planting obstacles [9]. The strip intercropped soybean enhanced the soybean yield without affecting the corn yield, which can balance high crop yield and sustainable agricultural development [10]. Nonetheless, in the maize–soybean strip intercropping system there were still some drawbacks, likewise, shorter crops (soybean) were shaded by the taller crops (maize) in the middle and later stage of soybean growth [11]. Maize shading caused the shade avoidance response for soybean, including slender stems and lower content of lignin in the stem, which in turn results in lodging to soybean [12–15]. Lodging was one of the important factors for the reduction of soybean yield and quality; studies have found that lodging causes soybean yield reduction up to 20–30% and up to 50% in severe cases [16,17]. Lodging mostly occurred in the internodes of soybean stem; therefore, the pros and cons of stem base traits were closely related to the lodging resistance of the plant [18–20]. An important indicator for measuring the degree of lodging resistance of soybean stems was the stem lodging resistance index associated with stem base resistance, stem thickness, and above ground fresh and dry biomass [16,21,22]. In maize–soybean relay intercropping, shading of maize increased the plant height and reduced the stem diameter of soybean [23]. Shading of taller crops to shorter crops and self-shading of crops with different densities reduced the lodging resistance index of soybean stems and increased the risk of lodging [16]. Previous studies have also shown that the contents of lignin and cellulose in the stem were closely related to the lodging resistance of plant stems [24–26]. The high activity of enzymes in the lignin synthesis pathway in soybean stem was beneficial to increased lignin content, which contributes to the resistance of plants to lodging [8,27–29].

The conversion of soluble sugars and sucrose in cotton was the premise of cellulose synthesis and the basis of cellulose formation [30]. Previous studies have proved that rice with high starch content in the stem has strong wind resistance [31]. Therefore, the higher contents of non-structural carbohydrates and structural carbohydrates in the stem were beneficial to the improvement of the lodging resistance of soybean. The contents of carbohydrate in soybean stems were closely related to leaf photosynthesis, photosynthetic products transportation from the source (leaf) to the sink (stalk) through the phloem sieve tube; thereby promoting the growth of the stem [32]. Plants require a particular intensity of light for their growth and development, as higher or lower light intensity than normal light intensity may stop photosynthesis. Under intercropping plants usually receive low light or shade stress as compared to monocropping with higher light intensities [33]. Low light or shade stress is the main chronic restraint that affects the crop growth under intercropping and high plant densities due to blockage of light [34]. It is described that due to decreased amount of photosynthetically active radiation (PAR) the light received by the plants, surroundings of plant i.e., carbon dioxide concentration, also affected that are necessary for plant growth [35]. Under low light or shade stress the activities of enzymes related to lignin synthesis are limited that caused low lignin content and lodging of plants [36]. Under the maize–soybean intercropping system, maize shades the soybean plants and significantly decreases the light transmittance to soybean canopy as compared to monocropping [23]. Different cultivation patterns have a great effect on plant photosynthesis; likewise, intercropping shading between different crops leads to a lower photosynthetic rate of shaded crops [37]. In multiple planting densities, mutual shading of different density plants leads to changes in the spatial structure of the plant population and photosynthetic rate, and ultimately the accumulation of carbohydrates in the stem that causes lodging easily [38–41]. At present, planting high density has already become one of the key measures to excavate the potential high yield of crops among the various cultivation methods [42,43]. Previous studies have found that planting density was significantly negatively correlated with the stem lodging resistance, excessively high planting density results in lodging, and lower grain yield, which seriously affected the normal growth and development of crops and yield [44–46]. Suitable field planting density could improve the structure of the plant population and create a better environment for plants to make

full use of light energy, promote photosynthesis, improve the lodging resistance of stem, and increase yield [47,48].

To sum up, soybean stem lodging was affected by many factors, most of the related studies focused on the seedling stage of soybean in maize–soybean relay intercropping system. However, the effect of changing the environment on the lodging resistance of soybean stem in the middle and later growth stages of the strip intercropping was unclear. This study mainly focused on the effects of multiple planting densities on the contents of structural carbohydrates and non-structural carbohydrates and metabolism of lignin in strip intercropping soybean stem. It was of great significance to determine how the lodging resistance index of strip intercropping soybean stems affected by multiple planting densities. To provide a scientific theoretical basis for improving maize–soybean cultivation measures and guide the optimum planting density that enhanced the lodging resistance and final yield of maize–soybean strip intercropping system.

## 2. Materials and Methods

### 2.1. Site Description

The two–year field experiment was carried out during 2019 and 2020 at the Chongzhou experimental farm of Sichuan Agricultural University (China, 103°39′ E, 30°33′ N). The soil of the experimental site has the following properties: organic matter content of 24.3 g kg$^{-1}$, total K of 15.2 g kg$^{-1}$, total N of 1.6 g kg$^{-1}$, total P of 1.3 g kg$^{-1}$, available K of 169.4 mg kg$^{-1}$, available N of 299.5 mg kg$^{-1}$, and available P of 36.5 mg kg$^{-1}$. The weather conditions of experimental site were shown in Figure 1b. The light transmittance of soybean canopy was measured by improved solar panel sensors (Figure 1a).

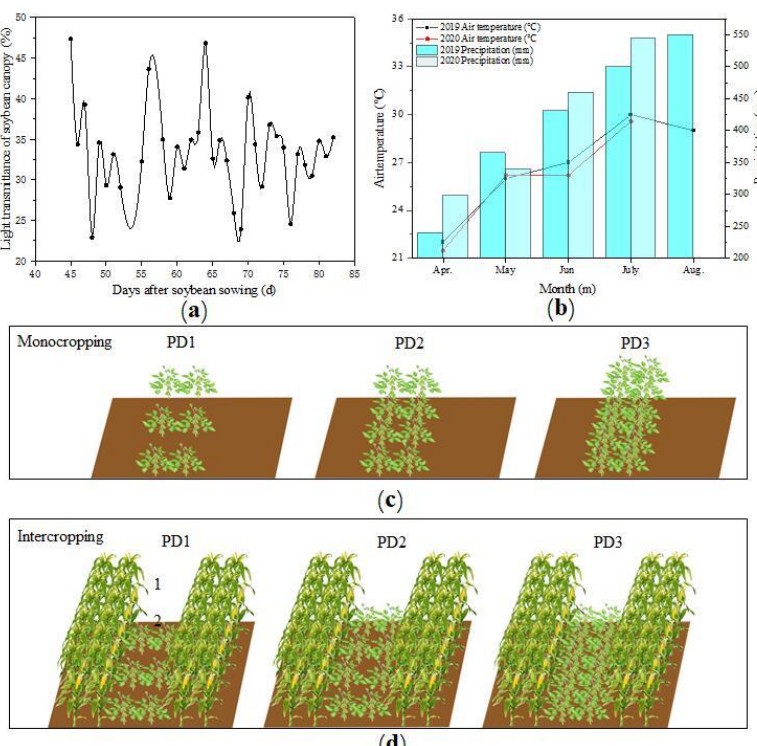

**Figure 1.** Multiple planting densities (PD1, PD2, and PD3) having a total number of 17 plants m$^{-2}$, 20 plants m$^{-2}$, and 25 plants m$^{-2}$, respectively, represented by (**c**) and (**d**) monoculture soybean and maize–soybean strip intercropping system. The dynamic change of light transmittance of soybean canopy from point 1 to point 2 showed in (**a**) and (**d**). The monthly average air temperature and precipitation from April to August were shown in figure (**b**).

## 2.2. Experimental Design

The field experiment adopted two-factor split design with three replications, and two plating patterns (maize–soybean strip intercropping and soybean monocropping) as the main factor and three planting densities (PD1, PD2, and PD3), (17 plants m$^{-2}$, 20 plants m$^{-2}$, and 25 plants m$^{-2}$), respectively, as the secondary factor. There were three strips of each treatment and one investigation strips was $6 \times 2$ m$^2$ and total experimental plot size 6 m$\times$ 6 m (36 m$^2$). The soybean cultivars Spring soybean (Chuandou$-$16) the main variety in southwest China and semi-compact maize (zhenghong505) were used in the experiment. In 2019 and 2020, both maize and soybean were sown on 2 April and harvested on 26 July at the same time. The maize strips were treated with compound fertilizer application at 800 kg ha$^{-1}$ (N:P: K = 15:15:15) as the base fertilizer before sowing. At the jointing and heading stages of maize, the urea (N $\geq$ 46%) was applied as fertilizer with 78 kg ha$^{-1}$ and 132 kg ha$^{-1}$ respectively [49]. However, there was no treatment in soybean strips.

The maize and soybean were planted in a 200 cm bandwidth with maize and soybean strips, including the row spacing of maize and soybean was 60 cm (Figure 1). In maize strip, the row spacing of maize was 40 cm, and the plant spacing was 20 cm, the maize planting was 10 plants·m$^{-2}$. Whereas in the soybean strip, the row spacing of the three densities for soybean was 40 cm, and the plant spacing of soybean was 8 cm, 10 cm, and 12 cm, respectively, which meant that the planting density treatments were 25, 20, and 17 plants·m$^{-2}$, respectively (Figure 1d). There was the same planting density but no maize strip for soybean monocropping as a control group (Figure 1c).

## 2.3. Data Collections

After 35 days of soybean sowing, the 10 soybean plants from each replication were sampled with 14 days interval until the final soybean harvest. Out of the 10 sampled plants, 5 plants were refrigerated at $-80$ °C for the determination of lignin related enzyme activities including phenylalanine ammonia-lyase (PAL), 4-coumaric acid: CoA-ligase (4CL), cinnamyl alcohol dehydrogenase (CAD), and peroxidase (POD), respectively. The other 5 plants were used for the measurement of stem morphological characteristics i.e., plant height, stem diameter, and stem folding resistance. After measuring, these plants were dried at 105 °C for 30 min, and then dried at 75 °C up to the constant weight for measuring carbohydrates contents including lignin content, cellulose content, sucrose, starch, and soluble sugar contents.

## 3. Sampling and Measurements

### 3.1. Plant Height and Stem Diameter

Fifteen uniform representative soybean plants were selected from each plot after 35, 49, 63, 77, 96 days of sowing. The stem diameter at the middle point of the first internode above the ground from the stem base of soybean was measured with a Vernier caliper. For the height of soybean plants, erect growth was considered as plant height, which was measured with a metric ruler.

### 3.2. Lodging Percentage

One strip in each plot of soybean was taken as the investigation zone, which was not sampled, and the survey time of the lodging percentage was consistent with the sampling time. The lodging percentage was measured as described by [50]:

Lodging Percentage (%) = total number of plant lodged in a plot/total number of plants in a plot $\times$ 100.

### 3.3. Stem Strength and Lodging Resistance Index

For stem strength, fifteen homogeneous soybean plants were taken from each treatment 35, 49, 63, 77, and 96 days after sowing (d). The stem strength was measured from the third internode to the fifth

internode with Digital Force Tester (YYD−1, Zhejiang Top Instrument Hangzhou, China). The plant height was measured with a metric ruler. The lodging resistance index was measured by the following formula [22], Lodging resistance index = Stem bending force ÷ (main stem length × above ground biomass fresh weight).

### 3.4. Evaluation of Stem Carbohydrates Content

For stem structural and non-structural carbohydrates evaluation fifteen uniform soybean plants were selected from each plot 35, 49, 63, 77, and 96 days after sowing. Stem samples were taken from the third internode to the fifth internode and grounded into powder and weighed 0.1 g to determine carbohydrates content. Then dried samples were ground into a powder with a Powder prototype (FW-80, Beijing), which can crush through an 80-mesh sieve to determine the following indicators.

### 3.5. Lignin Content

Lignin content was determined by using a test kit [51], (Lignin Content Test Kit, Qi Yi, Shanghai) with some modifications. There was a characteristic absorption peak at 280 nm after acetylation for the phenolic hydroxyl group in lignin, and the absorbance of 280 nm was positively correlated with lignin content. Dried samples of stem powder (2 mg) for each treatment were weighed into a 2.5 mL centrifuge tube, and 500 μL sulfuric acid and 20 μL perchloric acid were added. Then the tube was sealed with sealing film, fully mixed, and shaken once every 10 min in a water bath at 80 °C for 40 min, and then cooled at room temperature. From each sample 0.5 mL was transferred to a 50 mL centrifuge tube, then 995 μL NaOH solution added and fully mixed. For measuring the lignin value 200 μL from each sample solution was taken into a 96-well plate. The absorbance value of the blank tube and the measuring tube was determined at 280nm by the ELISA (SpectraMaxi3x from Austria). Lignin content was calculated according to the following formula.

$$\text{Lignin Content (mg/g)} = 0.075 \times (\Delta A - 0.0068) \div W \times T$$

where $\Delta A$; Light Absorbance value, W; Dry sample weight, and T; Dilution ratio.

### 3.6. Cellulose Content

The cellulose content was determined as reported earlier [13]. Then 0.3 g of a dried sample of grounded soybean stem was weighed into a 2.5 mL centrifuge tube. 1mL of 80% ethanol was added and homogenized the samples rapidly at room temperature and then the reaction was preceded at 95 °C for 20 min. The samples were cooled at room temperature and centrifuged 4000 rpm at 25 °C for 10 min and discard the supernatants. The precipitation was washed with 1.5 mL 80% ethanol and acetone (about 2 min by vortex oscillation and the supernatant was discarded by centrifugation 4000 rpm at 25 °C for 10 min). The precipitate was a coarse cell wall, soaked in 1 mL dimethyl sulfoxide solution (removing starch) for 15 h and centrifuged 4000 rpm at 25 °C for 10 min. Then the precipitation was dried, and the cell wall material (CWM) was weighed. Weighed about 5 mg of the dried CWM and added 0.5 mL distilled water to fully homogenize and transferred the homogenate to the tube. 0.5 mL distilled water added into the tube and put it into an ice bath. Then 0.75 mL concentrated sulfuric acid was added and kept for 30 min and then the samples were centrifuged for 10 min at 4 °C and 8000 rpm. After centrifugation samples were diluted with distilled water and 70 μL anthrone ethyl acetate and 630 μL concentrated sulfuric acid added and put it in a water bath at 95 °C for 10 min (cover tightly to prevent water loss). Cooldown at room temperature and absorbance value were measured by an ELISA at 620 nm. Cellulose content was determined by the following formula.

$$\text{Cellulose content (mg/g)} = 4.76\,(\Delta A + 0.0043) \div W$$

where $\Delta A$ = Light Absorbance value W; and Dry sample weight.

### 3.7. Determination of Soluble Sugar and Starch

According to the anthrone-sulfuric acid method [52], 100 mg dried samples of stem powder for each treatment were weighed into a 50 mL test tube. Then 1 mL of 80% ethanol was added, and the reaction was kept at 80 °C for 30 min. Then the samples were centrifuged at 6000 rpm for 10 min to collect the supernatant. This procedure was repeated twice. The collected supernatants of each sample were combined in a 100 mL beaker and placed in a water bath at 85 °C to evaporate the ethanol about 2–3 mL. After evaporation, the remaining liquid of each sample was transferred into a 50 mL volumetric bottle. Then 3 mL distilled water was added to the remaining precipitate and mixed, and kept in boiling water bath for 15 min. After cooling, 2 mL of cold 9.2 mol/l perchloric acid was added into each sample. Again 10 mL of distilled water was added into each sample and centrifuged 4000 rpm at 25 °C for 10 min. Then, the supernatant of each sample was transferred into a 50 mL volumetric bottle for the determination of soluble sugar and starch content. The absorbance soluble sugar and starch value was measured by an ELISA at 620 nm. Soluble sugar and starch contents were determined by the following formulas.

$$\text{Soluble sugar content (ug/g)} = AT \div W$$

where A = Content of soluble sugar in the standard curve, T; Dilution ratio, and W= Dry sample weight.

$$\text{Starch content (mg/g)} = 6.8A \div W$$

where A = Content of starch in the standard curve, and W = Dry sample weight.

### 3.8. Determination of Sucrose Content

According to the improved method for the determination of sucrose [53], dried soybean stem ground and sieved through 80 mesh size sieves. Then 0.1 g dried powder was weighed and put into a 50 mL test tube. Then 10 mL of 80% ethanol was added and heated in a water bath at 80 °C for 30 min. Then the supernatants were collected and centrifuged at 6000 rpm for 10 min. It was repeated twice and centrifuged in the same way. The three times collected supernatants were combined into 100 mL volumetric flask and the volume was fixed with distilled water. Then 1 mL of 0.1% phloroglucinol and 3 mL 10 mol/l HCL were added and placed in a water bath at 80 °C for 60 min. After cooling, the optical density was measured with a blank setting at the wavelength of 480 nm. Sucrose content was calculated according to the following formula.

$$\text{Sucrose content (mg/g)} = (A3 - A1) \div (A2 - A1) \div W$$

where A1 = Absorbance value of blank tube. A2 = Absorbance value of standard tube. A3 = Determine the absorbance value of the tube. W = Dry sample weight.

### 3.9. Key Enzymes of Lignin Synthesis Pathway

Fresh stem samples from the third internode to the fifth internode were grounded into powder and weighed 200 mg for lignin enzyme activity determination. The activity of lignin related enzymes i.e., PAL, 4CL, POD, and CAD were determined by micro determination (PAL, 4CL, POD, and CAD Test Kits, GeRuwasi, ShuZhou).

### 3.10. The Available Light of Soybean Plants

The available light in the canopy of soybean plants under different planting densities were measured with Plant Canopy Imager (CID, CI−110, US) after 35, 49, 63, 77, and 96 days of sowing. In all treatments, taking four evenly spaced measurements along a diagonal transect that runs between

two rows in every plot, and to do several transects to prevent the same plants from dominating the entire set of measurements.

### 3.11. Photosynthetic Rate of Soybean Inverted Trifoliate

For the measurement of the net photosynthetic rate, fifteen uniform plants were selected from each plot at V5 and R1of soybean. The photosynthetic rate of soybean inverted trifoliate was measured with a moveable photosynthetic instrument (Model LI-6400, LI-COR Inc., Lincoln, NE, USA). The instrument was calibrated as the following: photosynthetic photon flux density (PPFD) = 1000 μmol m$^{-2}$ s$^{-1}$, flow rate = 500 μmol mol$^{-1}$, stomatal ratio = 0.5, and ambient CO2 concentration = 400 μmol mol$^{-1}$. The aperture size of instrument was 6 cm$^2$ and the temperature of the leaf was recorded at 26 °C.

### 3.12. Statistical Analysis

All the data were collected statically analyzed by Origin Pro 2020b (OriginLab) software using one-way analysis of variance (ANOVA). Comparison among different treatments was based on Fisher LSD to assess the difference between treatments at probability levels of 0.001, 0.05, and 0.01. The figures were plotted by Origin pro 2020b and Microsoft Excel 2019. The asterisk showed a significant difference in different treatments. The Pearson correlation coefficient was used to determine the relationship between lodging resistance index and lodging rate carbohydrate contents and agronomic traits, the contents of carbohydrates of soybean stem, and its related enzymatic activity with a significance level of 0.05.

## 4. Results

### 4.1. Effect of Multiple Planting Densities on Lodging Resistance Index of Soybean

Our results exhibited that the lodging resistance index of soybean stem was significantly decreased under intercropping mode as compared to monocropping. In strip intercropping 35 days after sowing lodging was observed in PD3 and PD2 but in PD1 treatment 49 days after sowing lodging was observed. The lodging resistance index of soybean stem was significantly correlated with the increasing growth of maize-soybean and multiple planting densities. Likewise, in the strip intercropping 91 days after sowing the lodging percentage in PD2 and PD1 planting densities was decreased by 19.3% and 50.3%, respectively, than that of PD3 planting density treatment (Figure 2a). Our results showed that under multiple planting densities PD1, PD2, and PD3 the lodging resistance index of strip intercropped soybean was decreased by 15.4%, 22.7%, and 7.1%, respectively, as compared to mono-cropped soybean.

### 4.2. Effect of Multiple Planting Densities on the Available Light for Soybean Plants

In current findings, it was examined that under both (monocropping and strip intercropping systems the available light for soybean plants and maize-soybean growth and development were negatively correlated (Figure 3). Our results indicated that the available light for soybean plants in the strip intercropping was decreased by 38.5%, 33.6%, and 27.6% as compared to the monocropping under different planting densities PD1, PD2, and PD3, respectively. Moreover, in the strip intercropping mode the available light for soybean plants was decreased by increasing planting densities. Likewise, under multiple planting densities, the available light for soybean plants of PD3 and PD2 treatments was decreased by 28.5% and 13.7%, respectively, more than that of PD1 in the strip intercropping system.

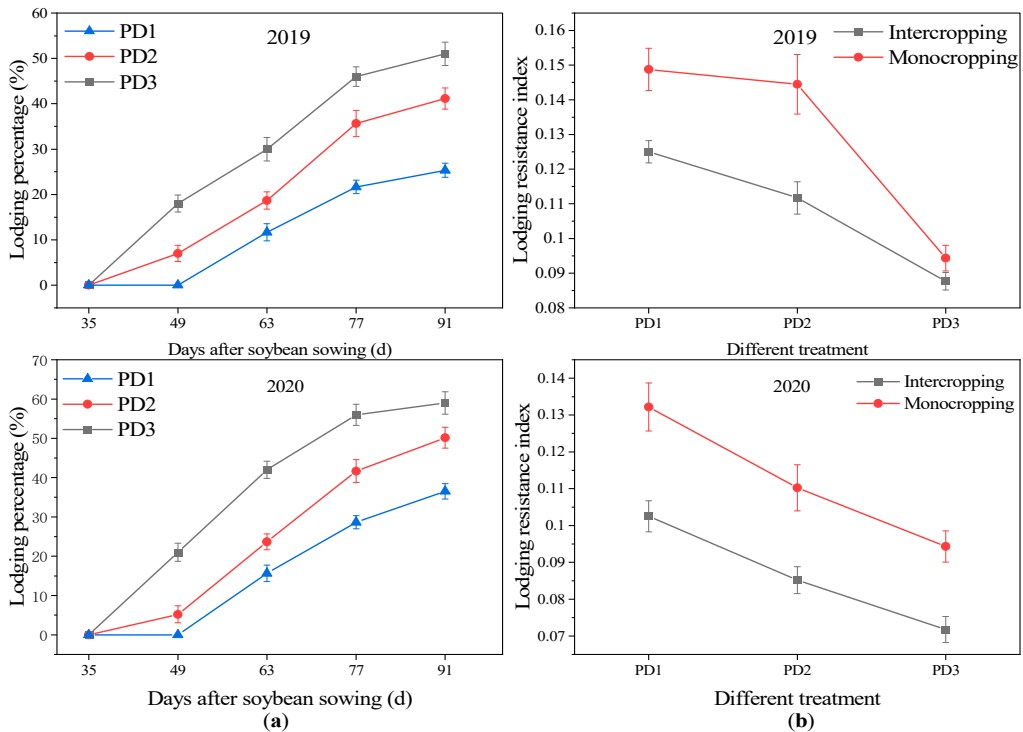

**Figure 2.** Effects of multiple planting densities (PD1 = 17, PD2 = 20, and PD3 = 25 plants m$^{-2}$) on the lodging percentage (**a**) and lodging resistance (**b**) under both monocropping and intercropping. All the values are means (±SE) standard error of three replications.

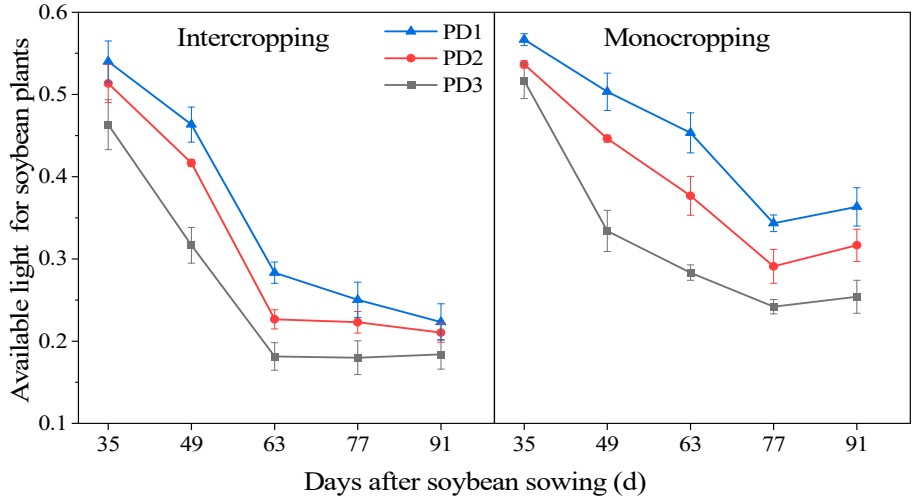

**Figure 3.** Effect of multiple planting densities (PD1 = 17, PD2 = 20, and PD3 = 25 plants m$^{-2}$) on the available light for soybean plants under both monocropping and strip intercropping system. All the values are means (±SE) standard error of three replications.

*4.3. Effect of MultiplePlanting Densities on the Net Photosynthetic Rate*

The net photosynthetic rate of soybean leaves at V5 and R1 growth stages was decreased gradually with the increase of planting density both in the strip intercropping and monocropping (Figure 4). Our study revealed that under both monocropping and strip intercropping system the average photosynthetic rate of soybean leaves at the R1 growth stage was highly significant under the PD3, PD2, and PD1 treatments, respectively (Figure 4b). However, at the V5 growth stage, the average photosynthetic rate of soybean had shown significance difference between different planting densities

treatments, but no significance differences shown under both the strip intercropping and monocropping system. Our findings revealed that PD1 showed the maximum net photosynthetic rate as compared to PD2 and PD3 treatments at R1 stage under both monocropping and strip intercropping. Furthermore, our results exhibited that, at the R1 growth stage, the average photosynthetic rate of strip intercropped soybean was enhanced by 10.5% and 15.2% of PD2 and PD1 planting densities, respectively, compared with PD3 planting density.

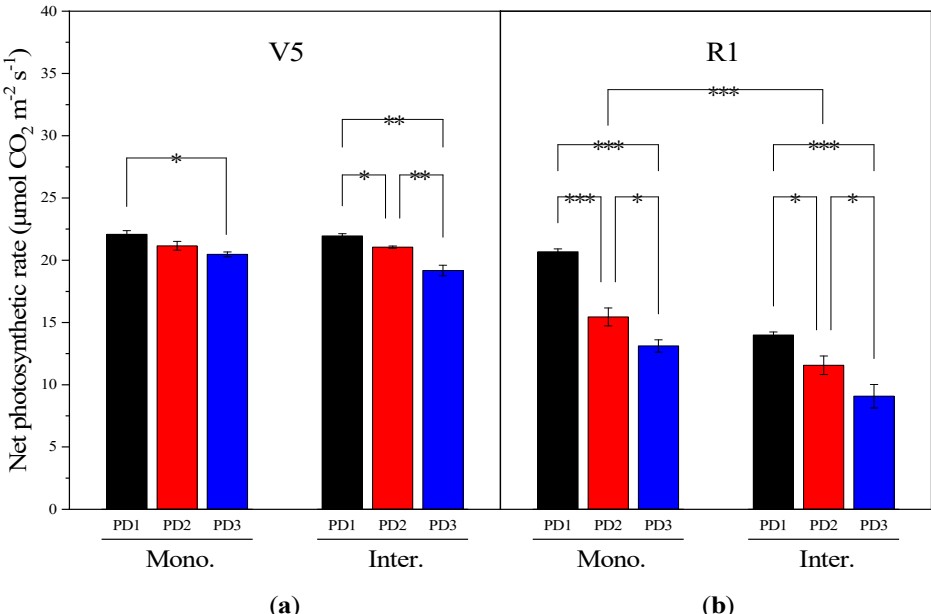

**Figure 4.** Effects of multiple planting densities (PD1 = 17, PD2 = 20, and PD3 = 25 plants m$^{-2}$) on the net photosynthetic rate of soybean leaves at V5 and R1 growth stages under both monocropping and strip intercropping system (**a**) and (**b**). All the values are means (±SE) standard error of three replications. Significance difference among the different treatments was displayed by *, **, and *** at probability level $p \leq 0.05$, $p \leq 0.01$, and $p \leq 0.001$, respectively.

### 4.4. Effects of Multiple Planting Densities on Non-Structural Carbohydrates Content of Soybean Stem

The contents of non-structural carbohydrates (soluble sugar, starch, and sucrose) gradually decreased with the increase of planting density both under the strip intercropping and monocropping (Figure 5). In the strip intercropping, for PD2 and PD3 treatments, the contents of soluble sugar and sucrose in stem gradually increased from 35 to 63 days after soybean sowing and decreased from 63 to 91 days after soybean sowing. While, for PD1 treatment in the strip intercropping, the contents of soluble sugar and sucrose in stem gradually increased from 35 to 77 days after soybean sowing and decreased from 77 to 91 days after soybean sowing. However, the changing trend of PD1, PD2, and PD3 treatments in monocropping was consistent with that of PD1 in strip intercropping. Under the strip intercropping, compared with PD3 treatment the content of soluble sugar in PD2 and PD1 treatments was increased by 10.1% and 79.8%, respectively. Likewise, for starch contents compared with PD3 treatment the contents in PD2 and PD1 treatments increased by 15.8% and 32.9% respectively. Furthermore, the starch content in stem gradually decreased from 35 to 49 days after soybean sowing and increased from 49 to 77 days after soybean sowing both in the strip intercropping and monocropping. Moreover, the content of sucrose in PD2 and PD1 treatments increased by 17.2% and 156.8%, respectively, as compared to PD3 treatment.

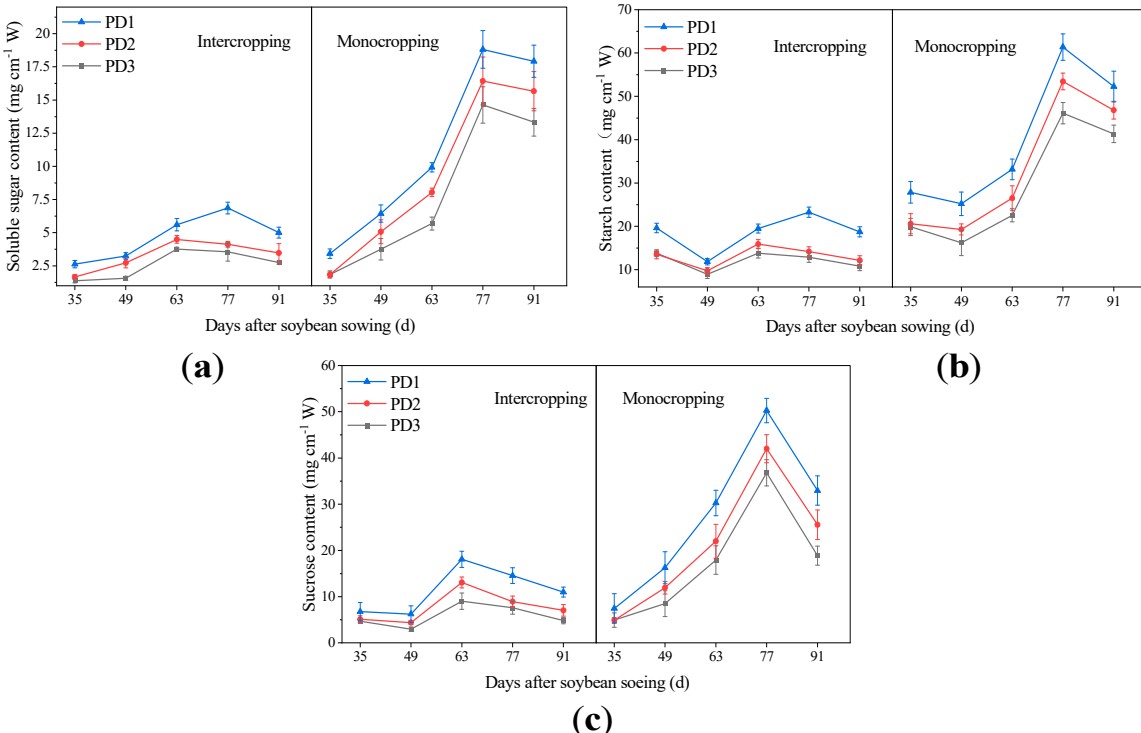

**Figure 5.** Effects of multiple planting densities (PD1 =17, PD2 =20, and PD3 =25 plants m$^{-2}$) on soluble sugars (**a**), starch (**b**), and sucrose (**c**) under both strip intercropping system and monocropping. All the values are means (±SE) standard error of three replications.

### 4.5. Effect of Multiple Planting Densities on Structural Carbohydrates Contents of Soybean Stem

Our findings concluded that multiple planting densities affected the contents of structural carbohydrates (cellulose and lignin) in the stem of strip intercropped and mono-cropped soybean significantly (Figure 6). However, the accumulation of structural carbohydrates in the two kinds of planting modes reduced significantly with increasing planting density. The accumulation of cellulose in PD2 and PD1 treatments was increased by 10.8% and 24.9%, respectively, under the strip intercropping as compared to PD3 treatment. While, compared with the monocropping, the accumulation of cellulose in PD3, PD2, and PD1 treatments decreased by 71.6%, 68.2%, and 65.7%, respectively, under strip intercropping. Furthermore, under the strip intercropping the accumulation of lignin in PD2 and PD1 treatments was enhanced by 18.5% and 36.7%, respectively, over PD3 treatment. Our findings revealed that under strip intercropping the accumulation of lignin in PD3, PD2, and PD1 treatments were decreased by 58.3%, 62.4%, and 60.5%, respectively, as compared to monocropping.

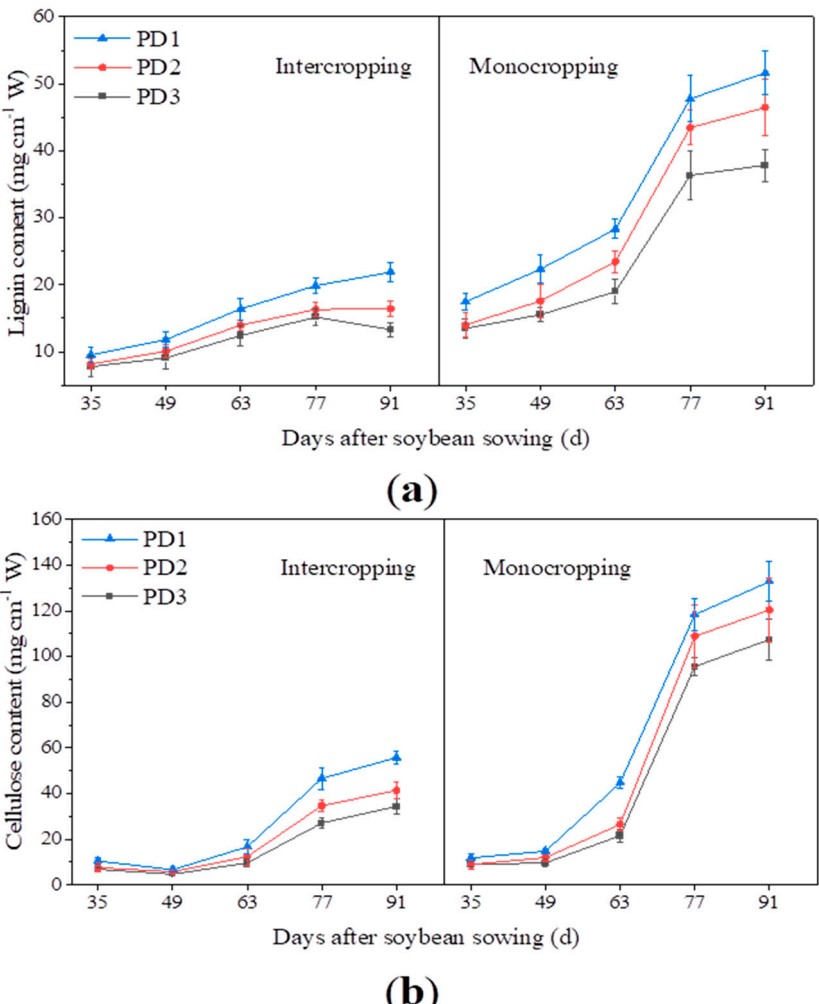

**Figure 6.** Effects of multiple planting densities (PD1 = 17, PD2 = 20, and PD3 = 25 plants m$^{-2}$) on lignin content (**a**) and cellulose content (**b**) under both strip intercropping system and monocropping. All the values are means (±SE) standard error of three replications.

### 4.6. Effect of Planting Densities on Key Enzymes of the Lignin Synthesis Pathway

Planting density significantly affected the activity of the key enzymes of the lignin synthesis pathway (PAL, POD, CAD, and 4-CL) both in the strip intercropping and the monocropping. As shown in Figure 7, the activity of enzymes decreased gradually with increasing planting densities. Our results exhibited that the activity of the PAL enzyme was enhanced by 11.7% and 27.1% in PD2 and PD1, respectively, under strip intercropping over PD3 treatment. Moreover, under strip intercropping the value of PAL enzyme in PD1, PD2, and PD3 was 66.4%, 69.1%, and 68.5 higher compared to the monocropping. Moreover, Figure 7b depicted that POD in each treatment gradually decreased from 35 to 49 days after soybean sowing and increased from 49 to 91 days after soybean sowing in the strip intercropping. However, in the monocropping, the activity of POD in PD1, PD2 gradually decreased from 77 to 91 days after soybean sowing; however, in PD3 treatment POD value from increased 77 to 91. The enzyme activity of POD in PD2 and PD1 treatments under the strip intercropping increased by 10.2% and 52.3%, respectively, as compared to PD3 treatment. In the same way, compared with the monocropping, the activity of POD in the PD3, PD2, and PD1 treatments in the strip intercropping was increased by 44.9%, 40.6%, and 61.6%, respectively.

However, the value of CAD showed an interesting trend (Figure 7c). Likewise, in each treatment, the CAD value first gradually decreased from 35 to 49 days after sowing. Then from 49 to 63 days

increased and then again decreased from 63 to 91 days after sowing under monocropping. However, PD1 treatment under strip intercropping also exhibited the same trend as monocropping. Furthermore, under strip intercropping, PD2 and PD3 treatment showed that the CAD value first gradually decreased from 35 to 49 days after sowing. Then from 49 to 77 days increased and then again decreased from 77 to 91 days after sowing. The activity of CAD in PD2 and PD1 treatments was increased by 6.1% and 23.8%, respectively, in the strip intercropping over PD3 treatment. Furthermore, our results indicated that the value of CAD under strip intercropping in PD1, PD2, and PD3 was enhanced by 16.3%, 17.1%, and 18.4%, respectively, as compared to the monocropping.

On the other hand, the activity of 4-CL in each treatment gradually decreased from 35 to 77 days and increased from 77 to 91 days after soybean sowing both in the strip intercropping and monocropping (Figure 7d). The activity of 4-CL in PD2 and PD1 treatments was 17.8% and 44.4% higher than that of PD3 treatment respectively under strip intercropping. Likewise, under strip intercropping the 4-CL enzyme activity in multiple densities PD1, PD2, and PD3 was enhanced by 55.3%, 0.7%, and 3.3%, respectively, compared with the monocropping.

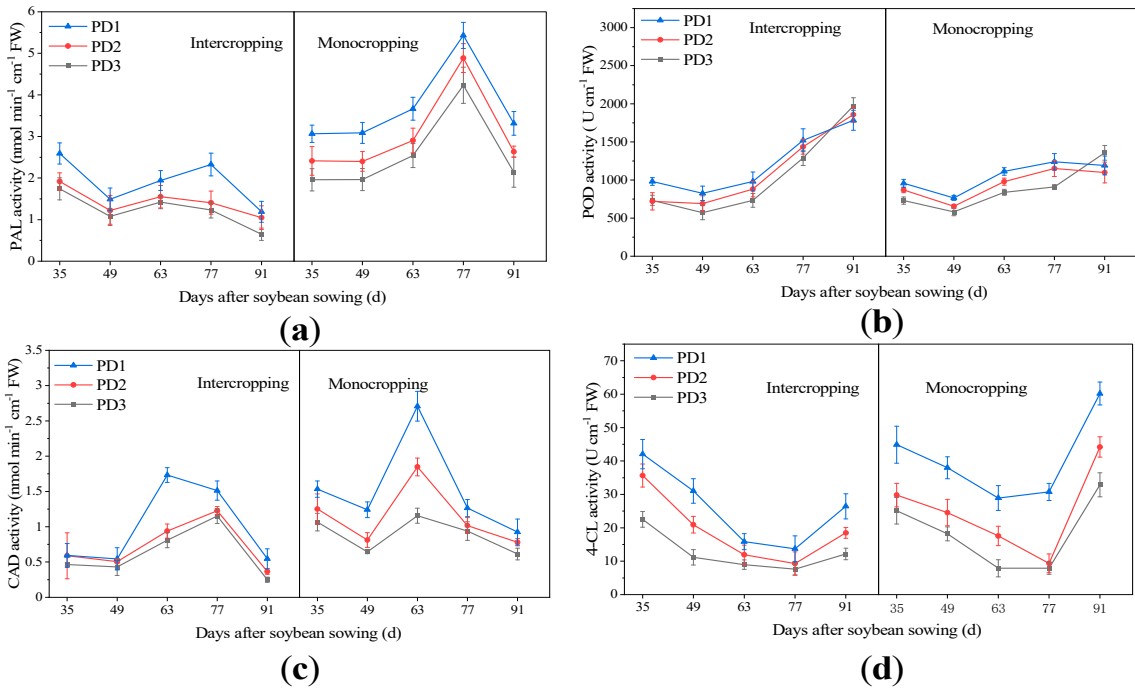

**Figure 7.** Effects of multiple planting densities (PD1 = 17, PD2 = 20, and PD3 = 25 plants m$^{-2}$) on PAL (**a**), POD (**b**), CAD (**c**), and 4CL (**d**) under both strip intercropping system and monocropping. All the values are means (±SE) standard error of three replications.

### 4.7. Effect of Planting Densities on Plant Height and Stem Diameter

The stem diameter and plant height of soybean gradually increased with the soybean growth and development both in the strip intercropping and monocropping (Figure 8). It can be seen in Figure 8a; the average stem diameter of soybean in the monocropping under the same density condition was higher than that of the strip intercropping. However, under the strip intercropping the stem diameter was 25.1%, 27.8%, and 23.7% lower than that of the monocropping under the PD3, PD2, and PD1 treatments, respectively. Our results revealed that within the increased planting density, the stem diameter of soybean got thinner, but the trend of the strip intercropping was greater than that of the monocropping. In the strip intercropping, compared to PD3 treatment, the stem diameter of PD2 and PD1 increased by 4.3% and 6%, respectively. In addition, the plant height in the strip intercropping was 28.1%, 31.8%, and 33% higher than that of the monocropping under the PD3, PD2, and PD1 treatments,

respectively. It was observed that within the increase of planting density, the plant height of soybean increased, but the increasing trend of strip intercropping was greater than that of the monocropping. Likewise, in the strip intercropping, the plant height of PD2 and PD1 reduced by 9.4% and 6.2%, respectively, as compared to PD3 treatment.

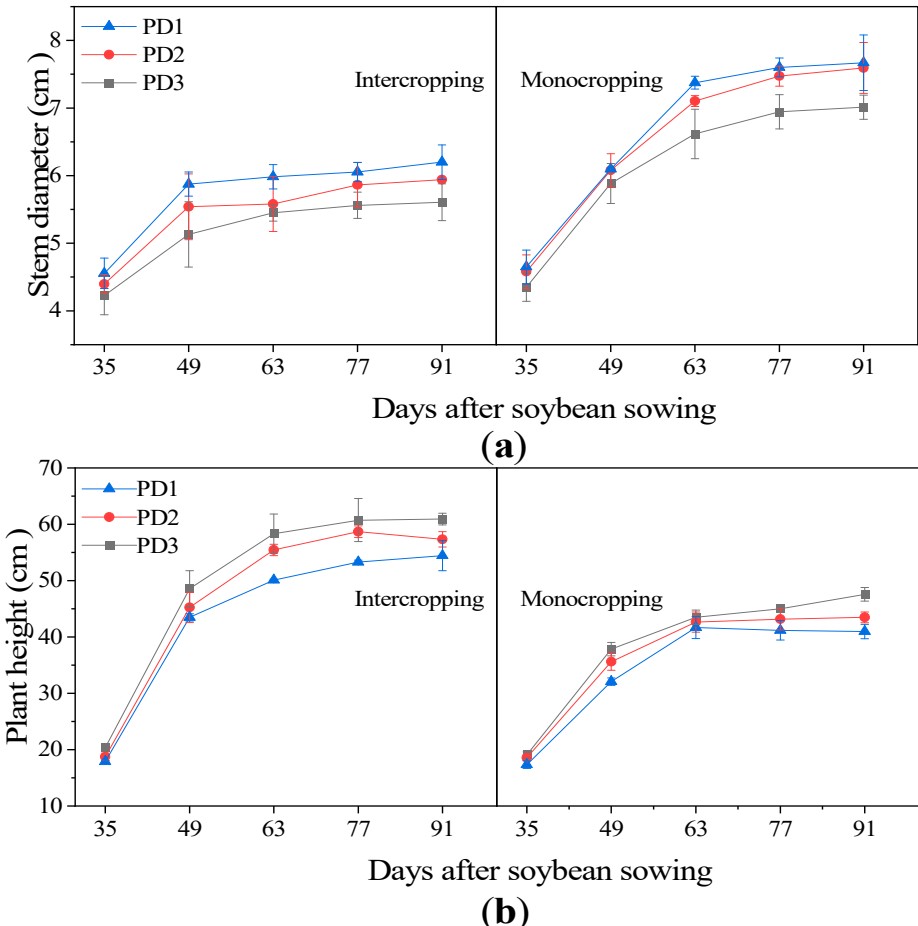

**Figure 8.** Effects of multiple planting densities (PD1 = 17, PD2 = 20, and PD3 = 25 plants m$^{-2}$) on stem diameter (**a**) and plant height (**b**) under both strip intercropping system and monocropping. All the values are means (±SE) standard error of three replications.

*4.8. The Contribution Rate of Carbohydrate Contents and Lignin Synthesis Related Enzyme Activities to Stem Strength*

Momentously, the stem strength/lignin content (SS/LC), stem strength/carbohydrates content (SS/CC), stem strength/sucrose content (SS/SUC), stem strength/soluble sugar content (SS/SSC), stem strength/starch content (SS/STC), stem strength/ cinnamyl alcohol dehydrogenase (SS/CAD), and stem strength/phenylalanine ammonia-lyase (SS/PAL) in strip intercropping were higher than that of monocropping, and they increased with the increase of planting density, although stem strength/peroxidase (SS/POD) showed the opposite trend (Figure 9). Compared with PD1 treatment, SS/LC, SS/CC, SS/SUC, SS/SSC, SS/STC, stem strength/4-coumaric acid: CoA-ligase (SS/4CL), SS/CAD and SS/PAL in stems under the PD2 and PD3 treatments was increased by 28.8%, 41.6%, 30.5%, 39.7%, 65.5%, 115.4%, 37.9%, 59.5%, 40%, 47.8%, 37.8%, 86.4%, 24.7%, 83.2%, 5.3% and 72.3%, respectively, in the strip intercropping. SS/POD in stems under the PD2 and PD3 treatments was decreased by 7.3% and 22.1%, respectively, compared with PD1 treatment.

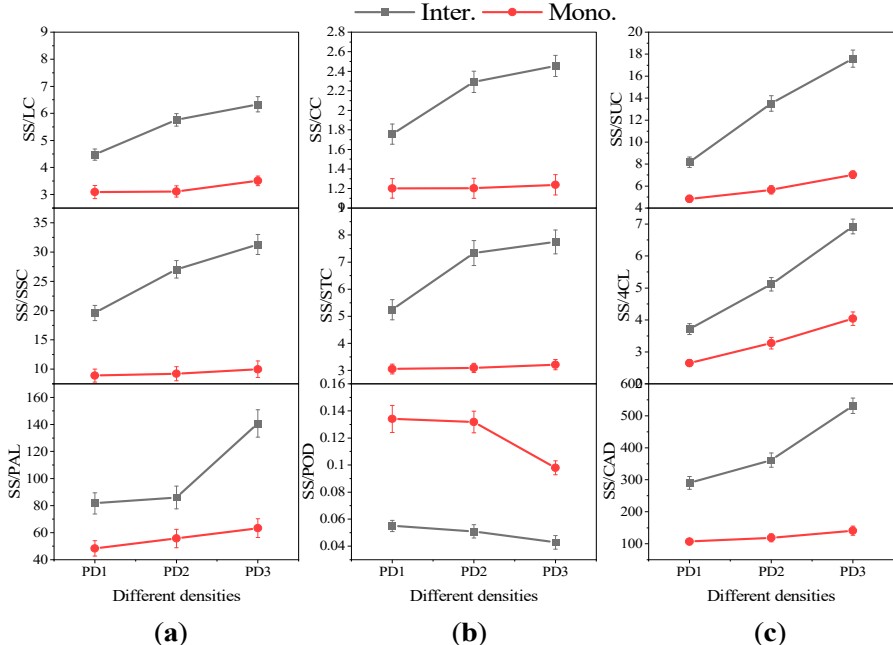

**Figure 9.** Contribution rate of lignin content (LC), soluble sugar content (SSC), and phenylalanine ammonia-lyase enzyme activity (PAL) (**a**); cellulose content (CC), starch content (STC), and peroxidase (POD) content (**b**); and sucrose content (SUC), 4-coumaric acid: CoA-ligase enzyme activity (4CL), and Cinnamyl alcohol dehydrogenase enzyme activity(CAD) (**c**) to stem strength (SS) under three different density treatments (PD1 = 17, PD2 = 20, and PD3 = 25 plants m$^{-2}$), respectively. All the values are means (±SE) standard error of three replications.

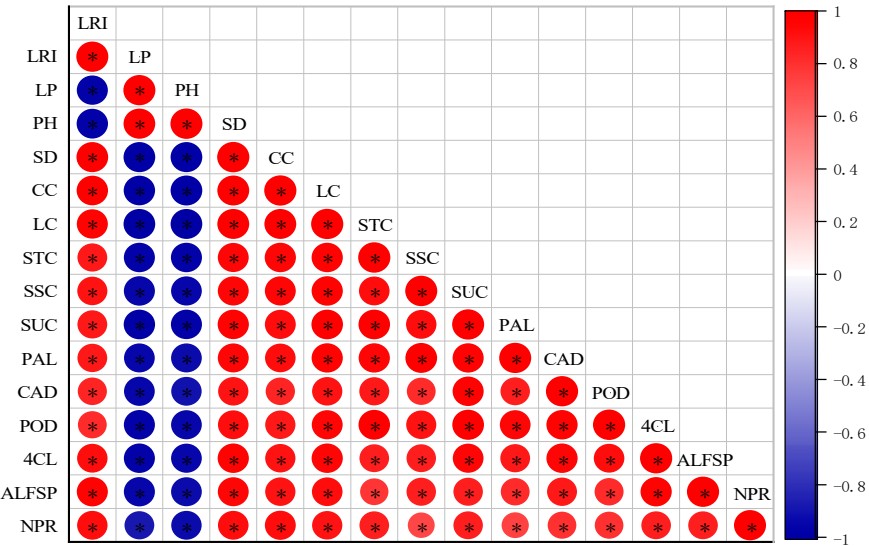

**Figure 10.** Correlation analysis of lodging resistance index of the stem (LRI) lodging percentage (LP), plant height (PH), stem diameter (SD), cellulose content (CC), lignin content (LC), starch content (STC), soluble sugar content (SSC), sucrose content (SUC), PAL enzyme activity (PAL), CAD enzyme activity(CAD), 4CL enzyme activity (4CLAvailable light for soybean plants (ALFSP), and net photosynthetic rate (NPR). The size of the circle represented the significance of the correlation, and the color of the circle represented the positive or negative correlation. The redder the circle is, the more significant positive correlation is, and the bluer the circle is, the more significant negative correlation is. The significant difference among the different treatments was displayed by * at probability level $p \leq 0.05$.

## 5. Correlation Analysis

As shown in Figure 10, the lodging resistance index of the strip intercropped soybean stem had significantly negative correlations with the field lodging rate and plant height of soybean. In the strip intercropping system, there were significant positive correlations between soybean stem lodging resistance index and structural carbohydrates contents (lignin and cellulose), non-structural carbohydrates contents (starch, sucrose, and soluble sugar), activities of enzymes related to lignin synthesis (PAL, CAD, POD, and 4CL), transmission coefficient, photosynthetic rate. (Figure 10).

## 6. Discussion

### 6.1. Light Environment and Photosynthesis Characteristics

In the maize–soybean intercropping system, the light was the most most-frequently limited factor under the condition of adequate water and the nutrient for crop growth and development [54–57]. In this study, there was a significant positive correlation between the net photosynthetic rate the available light for soybean plants. Previous studies have found that taller maize decreased the photosynthetic active radiation (PAR) reaching canopy of shorter soybean, increasing planting density of soybean increased the self-shading of crops, both of which lowered photosynthesis of soybean leaves [23,33]. It was observed that shading decreased the process of photo-inhibition in plants at midday and significantly enhanced the photochemical efficiency of photosystem II (PSII) [58]. Previous studies have shown that the available light for soybean plant was an important quantitative index of plant group structure, which directly reflects the size and shade degree of the population canopy [59,60]. Our findings revealed that the available light for soybean plants and photosynthetic rate of soybean leaves at V5 and R1 growth stage were lower in intercropping as compared to monocropping as shown in (Figures 3 and 4), with the increase of planting density, the available light for soybean plant, and the photosynthetic rate decreased gradually. Previous studies explained that the leaves of plants exposed to long-term low light radiation will become thinner, which weakened photosynthetic capacity [61]. It was also observed that different planting densities have an important effect on the structure and canopy of plants. Increasing the planting density caused the soybean group to cover each other, which reduced the available light for soybean plants. Eventually, the photosynthetic rate of the leaves was reduced [62,63]. However, what is noteworthy is that the exact pathways leading to a difference in the contribution of projection coefficient to net photosynthetic rate under different shading conditions are still unknown.

### 6.2. Lodging Resistance and Stem Morphology

The stem of soybean was an important part of supporting the whole plant, and lodging mostly occurred in the internodes of the stem; therefore, the pros and cons of the traits at the base of the stem were closely related to the lodging resistance of the soybean [64,65]. Previous studies have shown that the lodging resistance index of plant stem was strongly negatively correlated with the lodging rate [66,67], which is an agreement to our results as shown in (Figure 10). In the present study, compared with monocropping, the strip intercropping soybean with slender stems and higher plant height lodged seriously in the middle and later stages of soybean growth and the lodging resistance index of the stem were reduced as the increase of planting density as shown in (Figures 2 and 8). The reason may be that the strip intercropping soybean was shaded by maize in the middle and later stage, which led to the shade avoidance reaction occurred in soybean and caused stem lodging easily as depicted in (Figure 2a). It was reported that *Phellodendron amurense* seedlings adapt to shading by reducing stem diameter and seedlings biomass but increase plant height [68]. Additionally, it is also reported that sowing with appropriate density can reduce the lodging risk of maize and winter rapeseed [48,69]. Correlation analysis showed that the stem lodging resistance index was significantly positively correlated with stem diameter and remarkably negatively correlated with plant height under the different planting density as shown in (Figure 10).

### 6.3. Lodging Resistance and Carbohydrates Contents

Carbohydrates produced by photosynthesis of leaves were significantly positively correlated with the lodging resistance index of the stem, which was important substances involved in metabolism, and their distribution in plants was affected by both individual development and environmental factors [70–72]. The previous study has found that the Nandou12 (shade-resistance variety) with high contents of cellulose, soluble sugar, sucrose, and starch could increase the lodging resistance index of stems of intercropped soybean seedling [21]. Some studies demonstrated that high cellulose contents in stem could increase the physical strength of the stem and reduced the rate of lodging of soybean [13]. Our findings exhibited that the contents of cellulose, soluble sugar, sucrose, and starch in strip intercropping were so far below those in monocropping. Those contents in soybean stem showed a decreasing trend with the increase of planting density (Figure 5). This trend has also reported by previous studies that the soybean population structure was critically self-shading under high-density conditions, and the photosynthetic capacity was greatly affected, which was not conducive to the accumulation of carbohydrates in the stem [73–75]. As it can be observed in Figure 5 that the non-structural carbohydrates contents in stem showed the increasing trend and then decreased with the growth process of soybeans. Previous studies have shown that non-structural carbohydrates in the stems were not only providing energy to plants but also promote the accumulation of structural carbohydrates and partly transported to the seeds when the soybean was in the filling stage [38,39,76]. Our results showed that the starch, soluble sugar, and sucrose contents peak value time of PD1 treatment in strip intercropping were the same as that of monocropping; however, their contents were much higher than that of PD2 and PD3 treatments, which showed that a good light environment was beneficial to increase their contents. More importantly, in this study, the contribution rate of carbohydrate contents to stem strength (SS/CC, SS/SUC, SS/SSC, and SS/STC) in strip intercropping was far above that of monocropping (Figure 9). Furthermore, the contribution rate showed an increasing trend with the increase of planting density both in strip intercropping and monocropping. Previous studies have found that under low light conditions and drought stress significantly promoted the distribution of photosynthetic products from soybean leaves to stems [77,78]. Most studies proved that plants have evolved to adapt to shade, in which plants tend to reduce the leaf weight and distribute more photosynthate to the stem to enhance the stem strength [79,80]. Our findings revealed that under the strip intercropping, the contribution rate of carbohydrate contents to stem strength increased along with shading degree increase. However, the planting density (PD1) could appropriately decrease the self-shading among soybean plants as under PD1 distance between plants increased that provide the better light environment. Furthermore, under better light environment (PD1), the photosynthetic activity and carbohydrates content increased as compared to low light environment (PD2 and PD3) in strip intercropping soybean plants that ultimately enhanced the lodging resistance.

### 6.4. Lodging Resistance, Lignin Content, and Activities of Enzymes Related to Lignin Synthesis

Previous studies have demonstrated that the lodging resistance of stem was positively related to the content of lignin and activities of enzymes related to lignin synthesis. For instance, higher lignin accumulation at the base of the wheat stem could effectively increase the resistance to lodging and relieve the risk of lodging [81]. When the gene *RcPAL* up-regulated in the *Ricinus communis* L. stem, it remarkably increased the lignin content and dwarfed the plant height to increase lodging resistance [82]. In this study, the PAL, CAD, 4-CL, and POD enzyme activities were positively related to the content of lignin (Figure 10), which was beneficial to improve the lodging resistance of stem. It was reported that lignin was an important factor in the resistance of soybean seedling stems, the activity of POD, CAD, 4-CL, and PAL enzymes reflected the ability of stem resistance [8]. Correlation analysis showed that the activities of enzymes related to lignin synthesis in soybean stems were significantly positively correlated with stems lodging resistance index under the different planting density as depicted in (Figure 10). The findings of our study also revealed that lignin related enzymes were affected by maize shading in the middle and later stage of soybean, and with increased density,

those enzymes activity in the lignin synthesis pathway decreased (Figure 7). However, for the POD enzyme, as the density increased, its activity in soybean stems increased. Previous studies have found that increased hydrogen peroxide in the plants after soybean aging or drought stress induced an increase in POD enzyme activity, which in turn accelerates the lignification [83]. Our results indicated that the PAL and CAD enzyme peak value time of PD1 in strip intercropping was the same as that of monocropping. Furthermore, their activities were much higher than that of PD2 and PD3 treatments, which showed that a good light environment was beneficial to the synthesis of PAL and CAD enzyme. Similarly, the 4CL enzyme in strip intercropping was lower than that of monocropping, increasing planting density significantly decreased its activity (Figure 7d). However, the contribution rate of PAL, CAD, and 4-CL enzymes to stem strength (SS/4CL, SS/CAD, and SS/PAL) in strip intercropping was far above that of monocropping. The contribution rate showed an increasing trend with the increase of planting density both two planting patterns. While the contribution rate of POD enzyme to stem strength (SS/POD) was the opposite. Finally, in the maize–soybean strip intercropping, Suitable planting density (PD1) could create a good light environment, which was beneficial to increase the activities of lignin-related enzymes to improve stem lodging resistance.

## 7. Conclusions

In the strip maize–soybean intercropping system, the soybean growth is being affected by the spatial light pattern of this system. In the middle and later stages of strip intercropped soybean growth and development, shading of maize and self-shading between plants of soybean among different planting densities, significantly reduced the available light in the canopy of plant population and leaves photosynthetic capacity. However, the PD1 planting density significantly enhanced the photosynthetic activities, carbohydrates accumulation, and lignin and cellulose contents under strip intercropping system by providing a better light environment as compared to PD2 and PD3 planting densities. Correlation analysis also showed that the lodging resistance index of soybean stem and planting density was significantly correlated with the light environment, carbohydrates contents, and activities of enzymes related to lignin synthesis as compared to PD2 and PD3 planting densities. Moreover, our findings will be helpful to understand the mechanism of lodging resistance of intercropping soybean under different degrees of shading and also guide the optimum planting density. In future studies, the effect of multiple planting densities on lignin monomers, soybean stem anatomy, and secondary metabolism of intercropped soybean stem could be investigated to enhance the soybean lodging resistance.

**Author Contributions:** Conceptualization, W.L., W.Y.; Methodology, W.L., W.Y.; Formal Analysis, B.C., A.R; Investigation, W.L., W.Y.; Data Curation, B.C., A.R., L.W.; Original Draft Preparation, B.C., A.R., W.L.; Writing–Review & Editing, B.C., A.R., L.W., S.Q., M.X., I.A., T.Z., B.W., J.L., Y.G., Y.Z., All authors have read and agreed to the published version of the manuscript.

**Funding:** This work was funded by the National Natural Science Foundation of China (31201170) and Sichuan Innovation Team Project of National Modern Agricultural Industry Technology System, SCCXTD−2020−20.

**Conflicts of Interest:** The authors declare that they have no conflict of interest.

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
