# Peer review of "Effects of Multiple Planting Densities on Lignin Metabolism and Lodging Resistance of the Strip Intercropped Soybean Stem"

_agronomy, doi:10.3390/agronomy10081177_

Round 1

Reviewer 1 Report

Dear Authors,

This research is very interesting, and it has its scientific soundness, but some issues need addressing.

I hope my suggestions will be helpful!

Best wishes,

Reviewer

Review Agronomy:

Effects of multiple planting densities on lignin 2 metabolism and lodging resistance of the strip 3 intercropped soybean stem

L 2-4 Please capitalize all words in headings according to the language editing information in the Guidelines for authors.

L 30-31 Abstract should be max 200 words!

L 35-94 Please elaborate on why you included the transmission coefficient of soybean inverted trifoliate and the contribution rate of carbohydrate contents and lignin synthesis related enzyme activities to stem strength as parameters. What information do they give? Were they used in earlier lodging and shading studies?

L 96-101 Please include the information on the number of experimental unit replicates and years if existing. One-year, one-replicate experiments are not sufficient for drawing conclusions with certainty, particularly in matters influenced by the environment such as lodging. If this is the case, give sufficient explanation to justify such experimental design.

L 102-107 Please elaborate on the size/ area of the experimental plots. What is the length of the rows? The plot size is significant since you took 5 x 30 plants from each plot, and if plots were small, this could have had an impact on the measurements, observations and overall results as well.

L 111-115 Please elaborate on including the average canopy light transmittance diagram (Figure 1a) and its determination methodology, because ‘from point 1 to point 2’ (Figure 1c) is not clear enough.

L 118-120 Please include the abbreviations for mentioned parameters which you use later in L 382-389. According to the Guidelines for authors, abbreviations should be defined in parentheses the first time they appear in the text and tables or figures and used consistently thereafter.

L 130-134 Please elaborate on methodology as it is not very clear. What is the size of the investigation strip? The last sentence is not very clear as it doesn’t mean that lodging plants were those in which the stems were completely or partially broken or leaned 30° or more from the vertical and that the lodging percentage was calculated as the number of lodged plants in a plot∕ number of all plants in the plot × 100, as Da Bing et al. (2016) suggested. Please rephrase the sentence.

L 139 Are stem length and plant height the same? If not, what is the difference between these two measures?

L 143-210 Change ‘ml’ to ‘mL’ as indicated in the Guidelines for authors.

L 145-147 In L 118-122 you write that 15 plants per plot were taken for determination of LC, CC, SUC, SC, SSC, PAL, 4CL, CAD, and POD. In L 145-147 you write that parts of the stem from the third to the fifth internode of 15 plants were taken as samples for the determination of CH, dried to constant weight and ground into powder. Afterwards, in L 215-216 you write that you took parts of the stem from the third to the fifth internode of 15 plants and refrigerated these samples for enzyme determination. This doesn’t add up! Please revise. I assume that you took 15 plant samples and refrigerated them for enzyme determination and CH were determined on 15 plants you took as samples for determining stem morphology, but it is not clear.

L 219-226 Please elaborate on why you chose to take measurements on the inverted trifoliate?

L 225 Please elaborate on why you chose R4 and R6 for measuring photosynthetic rate.

L 267 You write that ‘The net photosynthetic rate of soybean leaves at V5 and R1 growth stages was decreased gradually…’ but in Materials and Methods in L 224-225 you wrote that measurements were taken in R4 and R6, and in Discussion (L 428-429) you mentioned R1 and R5. Which one is correct? Please revise!

L 269-271 The sentence is not really clear. Figure 4a indicates that there were differences between monocropping and intercropping, as well as differences between plant densities. Please rephrase.

L 271-273 The sentence is not really clear. What does ‘merely differences’ mean? Figure 4b indicates there were no differences between monocropping and intercropping, but there were differences between plant densities. Please rephrase.

L 273-275 Please rephrase this sentence.

L 275-277 You indicate that the increase of the photosynthetic rate between PD1 and PD3 (15.2 %), as well as between PD2 and PD3 (10.5%) can be seen from the Figure 4a, but that is not the case. The figures/ graphs depict only trends and approximate values. Please rephrase.

L 283-284 Earlier in the text, in the L 283-284, you wrote that the difference among the treatments was analysed at a probability level of 0.05, but here you have p<=0.05, p<=0.01, and p<=0.001. Please include these probabilities in the Materials and Methods.

L 328 Replace ‘and4-CL’ with ‘and 4-CL’.

L 333-337 Please revise and align with the results in Figure 7b. You write that ‘…POD in each treatment gradually decreased from 35 to 49 days after soybean sowing and increased from 49 to 91 days after soybean sowing in the strip intercropping and monocropping.’ but that is not true.

L 337 Replace ‘PD1treatments’ with ‘PD1 treatments’.

L 341-344 Please revise and align with the results in Figure 7c. You write that ‘…in each treatment, the CAD value first gradually decreased from 35 to 49 days after sowing. Then from 49 to 63 days increased and then again decreased from 63 to 91 days after sowing under both strips intercropping 343 (PD1) and monocropping.’ but that is not true.

L 348-350 Please remove the brackets from this sentence, and again as in L 275-277, the precise measure of increase between different values cannot be seen from the graph (see earlier comment).

L 367-369 Please rephrase the sentence. I assume you mean that within different cropping systems stem diameter got smaller as the plant density increased, but average stem diameter was bigger in intercropping than in monocropping for respective plant densities, but it is not clear.

L 382 See earlier comment for including the abbreviations for the parameters included in these ratios at first mention.

L 380-389 Stem strength should be written in full with the abbreviation in brackets as it is the first time it is mentioned. Furthermore, how did you measure this parameter? Why is this not included in the methodology? Is this the same as stem bending force? Please revise.

L 385-389 Again, the precise measure of increase between different values cannot be seen from the graph (see earlier comment), so please omit the ‘(Figure 9b)’ from the end of the sentence.

L 397-404 Parameters non-effective pods per plant, effective pods per plant and yield are not included in Figure 10, nor mentioned anywhere else in the text. Please revise.

L 423-426 Please rephrase this sentence, as it is not clear what you meant.

L 428-429 See earlier comment on time of photosynthetic rate measurement (V5 and R1, R4 and R6 or R1 and R5) and revise if needed.

L 442 Did you mean that lodging occurred in the basal internodes of the stem? Please revise.

L 475-478 Please rephrase these two sentences to avoid beginning the second sentence with an ‘and’.

L 482-484 Please rephrase this sentence for more clarity. I assume that you wanted to say that ‘Previous studies have found that low light conditions and drought stress caused a significant increase in the distribution of photosynthetic products from soybean leaves to stems.’ but it is not clear.

L 487-489 Please rephrase this sentence for more clarity. I assume you wanted to say that PD1 planting density could have lower self-shading of soybean plants which results in increased CH content and lodging resistance, but is not clear. If previous studies have found that low light conditions promoted the transfer of photosynthetic products from soybean leaves to stems, therefore increasing the contribution of CH to stem strength, what do you mean by ‘…the planting density (PD1) could appropriately decrease the self-shading among soybean and increase the carbohydrate content...’? As in the overall plant CH content in comparison to shaded plants?

L 500 Please replace ‘enzymes related to synthetic lignin in soybean stems’ with ‘enzymes related to lignin synthesis pathway in soybean stems’.

L 550 Please be consistent in the references considering the use of abbreviations of the journal names.

Author Response

Author’s response to reviewers

Reviewer #1:

  1. L 2-4Please capitalize all words in headings according to the language editing information in the Guidelines for authors.

Response:

First of all we would like to thank you for your appreciations and efforts toward our manuscript. We have modified our manuscript according to your suggestions. To see the modification please see the line numbers 2-4.

  1. L 30-31Abstract should be max 200 words!

Response:

We have modified the abstract section up to 200 words. To see the modification please visit the line numbers 13-36.

  1. L 35-94 Please elaborate on why you included the transmission coefficient of soybean inverted trifoliate and the contribution rate of carbohydrate contents and lignin synthesis related enzyme activities to stem strength as parameters. What information do they give? Were they used in earlier lodging and shading studies?

Response:

We have modified our manuscript according to your suggestions. To see the modification please see the line numbers 78-89. The contribution rate of carbohydrate contents and lignin synthesis related enzyme activities to stem strength described the stem strength over the amount of carbohydrates content deposited into stem by dividing the value of stem strength to carbohydrates content. As indicated in Figure 9 that under different planting densities it shows the trends and from this we can estimate the lodging under both intercropping and monocropping.

  1. L 96-101 Please include the information on the number of experimental unit replicates and years if existing. One-year, one-replicate experiments are not sufficient for drawing conclusions with certainty, particularly in matters influenced by the environment such as lodging. If this is the case, give sufficient explanation to justify such experimental design.

Response:

We have modified our manuscript according to your suggestions. To see the modification please see the line numbers 114, 115, 123, and 124. Two year field experiment with three replications was conducted during 2019 and 2020. We just showed lodging data of two year in the manuscript and other data were not shown yet.

  1. L 102-107 Please elaborate on the size/ area of the experimental plots. What is the length of the rows? The plot size is significant since you took 5 x 30 plants from each plot, and if plots were small, this could have had an impact on the measurements, observations and overall results as well.

Response:

We have modified our manuscript according to your suggestions. To see the modification please see the line numbers 123, 124, 126, and 127.

  1. L 111-115 Please elaborate on including the average canopy light transmittance diagram (Figure 1a) and its determination methodology, because ‘from point 1 to point 2’ (Figure 1c) is not clear enough.

Response:

We have modified our manuscript according to your suggestions. To see the modification please see the line numbers 119, 120, 140, 141, 148, and 149.

  1. L 118-120 Please include the abbreviations for mentioned parameters which you use later in L 382-389. According to the Guidelines for authors, abbreviations should be defined in parentheses the first time they appear in the text and tables or figures and used consistently thereafter. Response:

We have modified our manuscript according to your suggestions. To see the modification please see the line numbers 449-464.

  1. L 130-134 Please elaborate on methodology as it is not very clear. What is the size of the investigation strip? The last sentence is not very clear as it doesn’t mean that lodging plants were those in which the stems were completely or partially broken or leaned 30° or more from the vertical and that the lodging percentage was calculated as the number of lodged plants in a plot∕ number of all plants in the plot × 100, as Da Bing et al. (2016) suggested. Please rephrase the sentence.

Response:

We have modified our manuscript according to your suggestions. To see the modification please see the line numbers 127, and 169-174.

  1. L 139 Are stem length and plant height the same? If not, what is the difference between these two measures?

Response:

Yes, stem length and plant height are same. We replaced the word stem length with plant height, to see please see line number 179.

  1. L 143-210 Change ‘ml’ to ‘mL’ as indicated in the Guidelines for authors.

Response:

We have changed ‘ml’ to ‘mL’ according to your suggestions. To see the modification please see the line numbers 183-249.

  1. L 145-147 In L 118-122 you write that 15 plants per plot were taken for determination of LC, CC, SUC, SC, SSC, PAL, 4CL, CAD, and POD. In L 145-147 you write that parts of the stem from the third to the fifth internode of 15 plants were taken as samples for the determination of CH, dried to constant weight and ground into powder. Afterwards, in L 215-216 you write that you took parts of the stem from the third to the fifth internode of 15 plants and refrigerated these samples for enzyme determination. This doesn’t add up! Please revise. I assume that you took 15 plant samples and refrigerated them for enzyme determination and CH were determined on 15 plants you took as samples for determining stem morphology, but it is not clear.

Response:

We have modified our manuscript according to your suggestions. To see the modification please see the line numbers 186-188, 152-161, 256, and 257.

  1. L 219-226 Please elaborate on why you chose to take measurements on the inverted trifoliate?

Response:

We took measurements on the inverted trifoliate because it is functional leaf of soybean plant (outer leaf).

  1. L 225 Please elaborate on why you chose R4 and R6 for measuring photosynthetic rate.

Response:

We measured the photosynthetic rate at V5 and R1 stage (vegetative and reproductive stages of soybean) respectively. To see modifications please check the line number 271.

  1. L 267 You write that ‘The net photosynthetic rate of soybean leaves at V5 and R1 growth stages was decreased gradually…’ but in Materials and Methods in L 224-225 you wrote that measurements were taken in R4 and R6, and in Discussion (L 428-429) you mentioned R1 and R5. Which one is correct? Please revise!

Response:

We measured the photosynthetic rate at V5 and R1 stage. To see modifications please check the line number 271, 324, and 507.

  1. L 269-271 The sentence is not really clear. Figure 4a indicates that there were differences between monocropping and intercropping, as well as differences between plant densities. Please rephrase.

Response:

We have modified the sentence. To see modifications please check the line number 326-328.

  1. L 271-273 The sentence is not really clear. What does ‘merely differences’ mean? Figure 4b indicates there were no differences between monocropping and intercropping, but there were differences between plant densities. Please rephrase.

Response:

We have modified the sentence. To see modifications please check the line number 329-331.

  1. L 273-275 Please rephrase this sentence.

Response:

We have rephrased the sentence. To see modifications please see line numbers 331-335.

  1. L 275-277 You indicate that the increase of the photosynthetic rate between PD1 and PD3 (15.2 %), as well as between PD2 and PD3 (10.5%) can be seen from the Figure 4a, but that is not the case. The figures/ graphs depict only trends and approximate values. Please rephrase.

Response:

Thank you for your valuable suggestion. We rephrased these sentences all over the result section, to check please see the line numbers336-338, and 287-459.

  1. L 283-284 Earlier in the text, in the L 283-284, you wrote that the difference among the treatments was analysed at a probability level of 0.05, but here you have p<=0.05, p<=0.01, and p<=0.001. Please include these probabilities in the Materials and Methods.

Response:

We have added probability levels in the Materials and Methods section. To see please check the line numbers 280-282.

  1. L 328 Replace ‘and4-CL’ with ‘and 4-CL’.

Response:

We modified it. Please check line number 391.

  1. L 333-337 Please revise and align with the results in Figure 7b. You write that ‘…POD in each treatment gradually decreased from 35 to 49 days after soybean sowing and increased from 49 to 91 days after soybean sowing in the strip intercropping and monocropping.’ but that is not true.

Response:

We have revised and aligned the POD results according to Figure 7b. To see modifications please see the line numbers 394-396.

  1. L 337 Replace ‘PD1treatments’ with ‘PD1 treatments’.

Response:

We have replaced. Please see the line number 401.

  1. L 341-344 Please revise and align with the results in Figure 7c. You write that ‘…in each treatment, the CAD value first gradually decreased from 35 to 49 days after sowing. Then from 49 to 63 days increased and then again decreased from 63 to 91 days after sowing under both strips intercropping 343 (PD1) and monocropping.’ but that is not true.

Response:

We have revised the CAD results. To see please see the line numbers 406-410.

  1. L 348-350 Please remove the brackets from this sentence, and again as in L 275-277, the precise measure of increase between different values cannot be seen from the graph (see earlier comment).

Response:

We have modified the graph and remove brackets. To see modifications please see line numbers 413-415, and 336.

  1. L 367-369 Please rephrase the sentence. I assume you mean that within different cropping systems stem diameter got smaller as the plant density increased, but average stem diameter was bigger in intercropping than in monocropping for respective plant densities, but it is not clear.

Response:

We revised this sentence. To check please see the line number 432.

  1.  

L 382 See earlier comment for including the abbreviations for the parameters included in these ratios at first mention.

Response:

We have added the abbreviations. To see please see the line numbers 449-464.

  1. L 380-389 Stem strength should be written in full with the abbreviation in brackets as it is the first time it is mentioned. Furthermore, how did you measure this parameter? Why is this not included in the methodology? Is this the same as stem bending force? Please revise.

Response:

We have modified the manuscript according to your suggestions. To see modifications please see the line numbers 175-179. Yes, stem bending force and stem strength are same.

  1. L 385-389 Again, the precise measure of increase between different values cannot be seen from the graph (see earlier comment), so please omit the ‘(Figure 9b)’ from the end of the sentence.

Response:

We have omitted the ‘(Figure 9b)’. To see please check the line number 459.

  1. L 397-404 Parameters non-effective pods per plant, effective pods per plant and yield are not included in Figure 10, nor mentioned anywhere else in the text. Please revise.

Response:

As we did not mention the effective, non-effective pods per plant and yield data, so, we omitted these words from manuscript. To see modification please see the line numbers from 470,474, and 475.

  1. L 423-426 Please rephrase this sentence, as it is not clear what you meant.

Response:

The sentence was rephrased. To see please see the line numbers 498-500.

  1. L 428-429 See earlier comment on time of photosynthetic rate measurement (V5 and R1, R4 and R6 or R1 and R5) and revise if needed.

Response:

We have revised this sentence. To see modifications please see the line numbers 506-508.

  1. L 442 Did you mean that lodging occurred in the basal internodes of the stem? Please revise.

Response:

Yes, lodging of soybean start from base of stem 3rd to 4th internodes of the stem because due to shade the internode length was increased that cause early lodging from base of stem.

  1. L 475-478 Please rephrase these two sentences to avoid beginning the second sentence with an ‘and’.

Response:

Thank you for your valuable suggestion. We modified the sentence, to see please see the line number 553, and 555.

  1. L 482-484 Please rephrase this sentence for more clarity. I assume that you wanted to say that ‘Previous studies have found that low light conditions and drought stress caused a significant increase in the distribution of photosynthetic products from soybean leaves to stems.’ but it is not clear.

Response:

We revised this sentence. To see modification please see line numbers 562.

  1. L 487-489 Please rephrase this sentence for more clarity. I assume you wanted to say that PD1 planting density could have lower self-shading of soybean plants which results in increased CH content and lodging resistance, but is not clear. If previous studies have found that low light conditions promoted the transfer of photosynthetic products from soybean leaves to stems, therefore increasing the contribution of CH to stem strength, what do you mean by ‘…the planting density (PD1) could appropriately decrease the self-shading among soybean and increase the carbohydrate content...’? As in the overall plant CH content in comparison to shaded plants?

Response:

We have modified this sentence. To see please see the line number 566-571, and 98-101.

  1. L 500 Please replace ‘enzymes related to synthetic lignin in soybean stems’ with ‘enzymes related to lignin synthesis pathway in soybean stems’.

Response:

We have modified this sentence. To see please see the line number 583.

  1. L 550 Please be consistent in the references considering the use of abbreviations of the journal names.

Response:

We have revised the reference according to journal format. To see please see the reference section line number 635.

Reviewer 2 Report

The paper describes as plant density affects soybean plant growth and physiology, in particular lignin metabolism and lodging resistance of soybean in the intercropping system. I find the results very interesting aand useful. However the ms needs improvements, especially in methodologies. See specific comments in attached PDF.

Author Response

Author’s response to reviewers

Reviewer #2:

  1. L31: The authors should to emphasize the importance of this study.

Response:

First of all we would like to thank you for your appreciations and efforts toward our manuscript. We have modified our manuscript according to your suggestions. To see the modification please see the line numbers 33-36.

  1. L92: Why is it the great significance?

Response:

We have added the significance of this study. To see the modification please see the line numbers 107-109.

  1. L96: In which pweriod the experiment was carried out? How many years? Authors should show the meteorological trend, also as supplemental material.

Response:

The experiment was two year and carried out during 2019-2020. We have also added the two year meteorological data in paper. To see the modification please see the line numbers 114, 115, 118, and 119 (Figure 1b).

  1. L104: How was measured the light transmittance? To what plant density is the measurement referred?

Response:

The light transmittance was measured by solar panel sensor. As, our results indicated that PD1 plant density create a better light environment for soybean plants so, PD1 plant density can be referred, To see the modification please see the line numbers 119 and 120 (Figure 1a), and 26-30.

  1. L110: show PD1, PD2, and PD3

Response:

We have modified our manuscript according to your suggestions. To see the modification please see the line numbers 143 (Figure. 1).

  1. L115: In my opinion the light that arrives in 2 is also the incident PPFD and not all transmetted light! It should be the available light for soybean plants

Response:

We have modified our manuscript according to your suggestions. To see the modification please see the line numbers 146-150.

  1. L164, L183, L201 and L214: References for methods used.

Response:

We have modified our manuscript according to your suggestions. To see the modification please see the line numbers 205, 225, 243, and 191.

  1. L226: More details for Pan measurements: PPFD, temperature, CO2.....,

Response:

We have added more details about photosynthesis. To see the modification please see the line numbers 273-276.

  1. L229: Which statistical analysis was performed? Two-way Anova?

Response:

We used one-way ANOVA. To see the modification please see the line numbers 279 and 280.

  1. L262: "available light for soybean plants"..

Response:

We have modified our manuscript according to your suggestions. To see the modification please see the line numbers 318 (Figure. 3).

  1. L264: "available light....."

Response:

We have modified this line according to your suggestions. To see the modification please see the line numbers 319 and 320.

  1. L279: there is one "m" too many. show following the order: PD1, PD2, PD3

Response:

We have modified our manuscript according to your suggestions. To see the modification please see the line numbers 341 and (Figure. 4).

  1. L303 and L322: mg-1 cmW??????? are you shure?

Response:

We have modified our manuscript according to your suggestions. To see the modification please check the line numbers 365-366 (Figure. 5), and 385 (Figure. 6)

  1. L420: "available light for soybean plants"

Response:

We have modified this sentence according to your suggestions. To see the modification please see the line numbers 492, 493, and 607.

  1. L531: The authors should to emphasize the importance of this study.

Response:

We have added the importance of this study. To see the modification please see the line numbers 614-616

  1. L550: See the Intructions for Authors

Response:

We have modified our manuscript according journal format. To see the modification please see the Reference section, line number 635.

Round 2

Reviewer 2 Report

The manuscript has been improved.